# The Top Three Burning Questions in Total Hip Arthroplasty

**DOI:** 10.3390/medicina59040655

**Published:** 2023-03-26

**Authors:** Lefteris Manouras, Johannes Dominik Bastian, Nicholas Andreas Beckmann, Theodoros H. Tosounidis

**Affiliations:** 1Department of Orthopaedic Surgery, University Hospital Heraklion, Crete, 71500 Heraklion, Greece; 2Department of Orthopaedic Surgery and Traumatology, Inselspital, Bern University Hospital, University of Bern, 3010 Bern, Switzerland; 3Department of Orthopaedic Surgery, Center for Orthopaedics, Traumatology and Spinal Cord Injury, Heidelberg University Hospital, 69118 Heidelberg, Germany

**Keywords:** total hip arthroplasty, spinopelvic mobility, fast-track protocol, robotic systems, navigation systems

## Abstract

Total hip arthroplasty (THA) for end-stage osteoarthritis is one of the most effective surgical treatments in medicine. Impressive outcomes have been well documented in the literature with patients gaining ambulation and recovery of hip joint function. Nevertheless, there are still debatable issues and controversies that the orthopedic community has not been able to provide a definitive answer for. This review is focused on the current three most debatable issues surrounding the THA procedure: (1) new cutting-edge technology, (2) spinopelvic mobility, and (3) fast-track protocols. The scope of the herein narrative review is to analyze the debatable issues surrounding the three aforementioned topics and conclude the best contemporary clinical approaches regarding each issue.

## 1. Introduction

The first publication pertaining to total hip arthroplasty dates back to the 1940s [1,2]. Twenty years later, the concept of low-friction total hip arthroplasty in the United Kingdom was developed by Sir John Charnley (THA) [3,4]. Despite the fact that the initial outcomes of THA were rather disappointing, there have been significant advances over the last three decades that have led to significant improvements and, consequently, better clinical outcomes [5]. Hip arthroplasty is estimated to have a 95% survivorship at a 10-year follow-up (The NJR Editorial Board, 2013). Nowadays, THA is considered a tried-and-true milestone surgery and the clinical outcomes of the last decades have documented that it is one of the most effective and successful operations throughout all fields of surgery [6].

Despite the proven effectiveness of THA, there are still unsolved controversies and debates in the field. In the herein narrative review, we attempt to summarize the contemporary questions on three important issues pertaining to THA. Three major questions were arbitrarily chosen based on the clinical experience and everyday clinical of the leading authors of this study. Consequently, we tried to answer the following questions: (1) “How are the outcomes of robotic-assisted THA versus navigated versus conventional THA, are compared?” (2) “What is the correlation between spinal and pelvic mobility (spinopelvic tilt) and how it affects THA outcomes?”, and finally (3) “What is the current knowledge pertaining to the fast-track protocols in THA?”

## 2. Significance of Present Review: Discussing three Burning Topics/Questions in THA

Are new technologies (robotic or navigated systems) better than conventional THA?How important is spinopelvic mobility for THA?Is fast-track THA surgery beneficial and who is the ideal candidate for it?

## 3. Device-Assisted THA Versus Conventional THA

Device-assisted THA includes all new technologies which help surgeons to be more accurate in prosthesis positioning during hip arthroplasty surgery. These technologies are generally divided into robotic and computer-assisted navigation systems and are briefly depicted in Figure 1. 

One of the first introductions of robotic total hip arthroplasty (RO THA) dates back to the 1990s [7]. These early robotic systems were fully active, and the femoral osteotomy as well as the implant could be performed independently by the surgeon. Nevertheless, a number of limitations were associated with the first-generation fully active robotic systems, namely soft tissue trauma, periprosthetic fractures of the femur, and high conversion rates to conventional THA [7,8]. To address these issues, a second generation of haptic/semiactive robotic systems followed. As their name implies, these systems are not completely automated, but the surgeon controls the femoral neck cut, the acetabular preparation, and the position of the cup and the stem through a robotic arm-assisted system. A preoperative CT scan and subsequent 3D reconstruction are typically needed. A recent in vitro cadaveric study [9] utilized a coordinate measurement machine (CMM), utilizing specific landmarks of the pelvis to set up a digitalized anatomical coordinate frame. Using plain anteroposterior radiographs and a mathematical formula, the 3D anteversion and inclination of the acetabular cup were estimated. According to this study, based on 2D fluoroscopy images, a surgeon can calculate 3D angles for anteversion and inclination. This method is proposed to improve the acetabular component intraoperatively [9].

**A.** 
**Robotic Systems in THA**


Increased accuracy of implant positioning and less intraoperative trauma with subsequently fewer complications are factors that may improve THA survivorship through the use of robotic systems. This does not necessarily translate to increased clinical benefit; therefore, patient-reported outcomes (PROs) must be validated for an overall evaluation of robotic surgery.

Several studies and meta-analyses have been conducted in order to compare RO THA to conventional THA. In 2010, Nakamura et al. conducted a study comparing the outcomes and complications of traditional, i.e., hand-rasping implantation to active robotic-assisted techniques [10]. The authors concluded that more precise robotic-assisted implant positioning resulted in decreased led length discrepancy variance and less proximal femoral stress shielding five years after surgery. After ten years, the same group in a follow-up study demonstrated no clinical and/or radiographic advantages of robotic-assisted stem implantation compared to hand rasping during THA, concluding no benefit of robotic use [11]. On the other hand, Bargar et al. in a long-term 14-year follow-up found that patients with robotic-assisted THA scored higher for several PROs [12].

The aforementioned studies took into consideration only fully active robotic systems for THA that are not widely used in contemporary practice. Chen et al. conducted a meta-analysis of 7 studies involving 1516 patients to compare fully active or semiactive robotic surgery to traditional THA [13]. According to their findings, robotic-assisted THA was associated with lower intraoperative complication rates, improved acetabular cup and femoral stem positioning, as well as better global offset. On the other hand, a higher rate of heterotopic ossification was documented, attributed possibly to locator pin placement. Nevertheless, no difference in functional scores, limb length discrepancy, stress shielding, and overall revision rate were documented. Based on the aforementioned findings, the authors of the meta-analysis reported that robotic-assisted THA has certain benefits compared to conventional THA.

**B.** 
**Semiactive Robotic Systems**


Studies comparing semiactive robotic to manual THA have shown more promising results. Notably, Domb et al. [14] and Bukowski et al. [15] documented better postoperative PROs for semiactive robotics. To the best of our knowledge, the former study is the only one in the contemporary literature that demonstrates medium-term functional outcomes with a significant difference in PROs at 5 years. At the same time, these studies reported low complication rates for both conventional THA and semiactive robotic THA.

An overview of meta-analyses published in 2021 suggested that the accuracy of component positioning could be improved, and the number of intraoperative complications could be reduced with the utilization of robot-assisted THA [16]. On the other hand, the findings affirmed that robot-assisted THA duration is prolonged by 20 min, and that robot-assisted THA is associated with increased risks of postoperative heterotopic ossification, dislocation, and revision. However, no substantial differences in clinical or functional scores between robot-assisted and conventional THA were found.

A recent study from 2022 sought to evaluate dislocation rates and related revisions, as well as cup position, PROs, and postoperative complications when comparing RO THA and manual THA [17]. The study included a total of 2247 patients who received a primary semiactive robotic-assisted THA (523) or manual THA (1724) performed via posterior-based approaches. The study resulted in lower rates of dislocation, greater cup anteversion, and lower cup inclination in the RO THA patients, while no meaningful differences were documented for demographics, PROs, or additional complications. Furthermore, in another recent study, semiactive robot-assisted THA was reported to provide more accurate acetabular cup prosthesis implantation but was associated with longer operation times compared to conventional THA [18].

The opinion of the authors of this review is that in the future, studies and reviews should focus on making a discrete distinction between active and semi-active robotic assistance, addressing technology matureness, and considering surgeon experience. Clear limitations of RO THA include the additional radiation exposure and associated hazards and the high costs related to the installation, computer software, pre-operative imaging, operational costs, surgical team training, learning curves, and the limited amount of compatible implants [19,20]. However, it has been suggested that additional costs could be balanced by the reduced length of stay, fewer complications, and longer implant survivorship [21,22]. The novelty of robotics within THA has been associated only with short-term and medium-term follow-up outcomes. Therefore it is of paramount importance that well-designed long-term follow-up studies are conducted in the near future. Lastly, longer-term follow-up studies are required to assess improvement in outcomes [6] (Learmonth et al., 2007).

**C.** 
**Navigation Systems in THA**


Apart from the robotic systems, computer-assisted navigation systems have also been used in THA [23], and their use has been associated with improved precision in the acetabular component positioning. Computer-assisted navigation systems exist in two separate forms: those relying on pre-operative imaging and ‘imageless’ systems. ‘Image-driven’ systems rely on images produced by pre-operatively computer tomography (CT) scans. ‘Imageless’ systems entail that anatomic feature recognition is taking place intra-operatively and this information is used to create the surgical navigation plan. Despite the fact that CT-based navigation has been highly accurate, it is related to increased costs, the necessity for specific pre-operative imaging, and the unavoidable radiation exposure risk [24]. A large cohort study, based on a single-surgeon data series, compared an imageless navigation system with CT-based navigation systems [25]. This study sought to determine the intraoperative cup positioning compared to the universally accepted target values of 40° of inclination and 15° of anteversion, and at the same to appraise the accuracy, precision, and robustness of the navigation system. The authors concluded that the accuracy and precision of cup positioning were inferior compared to CT-based navigation systems. Image-less navigation provides a perioperative tool to achieve sufficient placement of the acetabular components [25].

One of the first meta-analyses in this field reported that the use of navigation in hip replacement compared to freehand THA improves the precision of acetabular cup positioning by decreasing the number of outliers from the anticipated alignment [26]. A major limitation of the study was the mixing of image-driven and imageless navigation THAs were combined into one group for analysis. In another study, Lass et al. [27] demonstrated that imageless navigation is helpful for the accurate positioning of the acetabular component [27]. A minimum 2-year follow-up of the aforementioned prospective randomized study was published in 2020. It concluded that imageless hip navigation indeed increases the accuracy of acetabular component placement within the target position and safe zones in comparison to conventional freehand implantation techniques. Nevertheless, the authors advocated that prospective randomized studies with long-term follow-up are needed to investigate the association of improved cup positioning to clinically meaningful outcomes [28]. Imageless navigation systems have also been found to offer more accurate anteversion and inclination positioning for acetabular cups than freehand techniques [29]. Additionally, one systematic review concluded that alternative methods for THA (computer navigation system (CAS), Optimized Positioning System (OPS™), and manual instrumentation) are safe and effective and result in acceptable acetabular cup positioning on postoperative imaging [30].

An imageless navigation systems is a computer-assisted procedure, where the surgeon uses three-dimensional sensors to detect anatomical features. An important meta-analysis published later in 2022 included all the clinical trials comparing imageless navigation versus conventional primary THA [31]. The analysis consisted of 21 studies (2706 procedures) aiming to compare cup inclination and anteversion, leg length discrepancy, surgical duration, Harris Hip Score, and rate of dislocation between the two THA approaches. Compared with conventional THA, the navigated group demonstrated slightly lower leg length discrepancy but a longer duration of the surgical procedure. The cup anteversion and inclination, Harris Hip Score, and rate of dislocation were similar between the two interventions. This latter study suggests that imageless hip navigation has not yet been widely accepted. More consolidated scientific validation is required to confirm its implementation in everyday clinical practice.

Overall, robotic and navigation technologies, can offer individualized planning and positioning of the acetabular cup, utilizing patient-specific safe zones of positioning modeled on preoperative assessment spinopelvic mobility. Nonetheless, the correlation of improved early functional outcomes to increased accuracy of implant positioning and targeted alignment utilizing these new technologies has not been documented up to date [32]. Moreover, it has been demonstrated that although robotic THA increases the accuracy in cup positioning, there is no difference in dislocation and revision rates between robotic and conventional THA [33]. At the same time, a recent review concludes that more accurate component positioning and restoration of the center of rotation results from the use of RO THA [34]. However, this improvement in radiological parameters of more accurate cup positioning has not been translated into better PROMs and/or improvement in leg length inequality. Consequently, additional research is warranted in order to investigate the long-term outcomes of each one of the aforementioned intraoperative modalities before definitive conclusions are drawn in regard to their pragmatic clinical efficacy.

## 4. Spinopelvic Mobility

Spinopelvic mobility embodies the dynamic and complex biomechanical interaction of the spine, pelvis, and hip, which is fundamental for normal human movement and posture. Understanding this interaction is of paramount importance for both hip arthroplasty and spine surgeons. In spinopelvic motion from standing to sitting, prevention of impingement of the femoral neck to the anterior acetabulum with resultant posterior dislocation is prevented with the normal decrease in lumbar lordosis, flexion of the hip, and posterior sacral tilt (rollback of the pelvis) [35]. When changing position from standing to supine, the anterior move of the pelvis moves results in less acetabular anteversion [36]. According to Phan et al. [37], there are four different types of spinopelvic mobility based on spinopelvic pathology: (1) flexible and balanced, (2) flexible and unbalanced, (3) rigid and balanced, and finally (4) rigid and unbalanced. In the same study, the authors suggest that treatment and especially cup positioning should be embedded according to the type of spinopelvic mobility. Lum et al. [35] also suggest that there are some patients that could benefit from dual mobility articulation arthroplasty. These patients are those whose ante-inclination (AI) measurement changes < 5° between sitting and standing positions, and those whose AI values are >75° in the sitting position.

Spinopelvic mobility can be affected by degenerative diseases of the hip and spine, as well as previous spinal fusion. In the geriatric population, there is an increase in degenerative musculoskeletal diseases and, as anticipated, an increase in patients with concomitant degenerative spine and hip pathological conditions. Consequently, the population of patients demanding lumbar fusion and THA will also increase considerably [38,39,40]. The coexistence of hip osteoarthritis and adult spinal deformity is very common, leading to sagittal spinopelvic malalignment. It is not clear which of these two pathologies must be corrected first [41,42]. However, the functional orientation of the acetabulum can be affected by alterations in spinopelvic mobility. To prevent complications after THA, it is important to preoperatively focus on the interplay between spinopelvic mobility parameters and the positioning of the implant [43]. For example, in a stiff spine, posterior dislocation of a THA can result when the stiff spine leads to a loss of the normal posterior pelvic tilt and anterior impingement when sitting.

The terminology used to describe the understanding and the various parameters of spinopelvic mobility and its implications in cup positioning has been confusing. Therefore a Hip-Spine Workgroup was created at the American Academy of Orthopedic Surgeons Annual Meeting in 2018 to facilitate the standardization of the terminology used to describe spinopelvic disease [44]. It is important to note that due to the progressive degeneration process, imbalance and spinopelvic mobility may deteriorate over time. Consequently, it becomes clear that spinopelvic mobility potentially a dynamic condition, and thus the classification of patients may change over time.

Dislocation is a common complication after THA [45]. Currently, it is well-documented that an increased risk of dislocation exists, even in patients where the acetabular component has been placed within the traditional “safe zone” [46,47], as described first by Lewinnek et al. [48] due to abnormal spinopelvic motion from lumbar pathology. Spinopelvic abnormalities are documented in very high numbers (75% to 92%) of THAs that dislocate, even when other possible causes of instability are identified [49,50]. Furthermore, it is now well-documented that spinal fusion prior to THA increases dislocation rates [51,52,53,54]. Importantly, recently published evidence is suggestive that spinal fusion surgery is the single strongest predictor of hip instability, even when compared to Parkinson’s disease, dementia, and psychiatric illness [55]. A meta-analysis inclusive of six studies revealed a twofold increased risk of dislocation and a threefold increased risk for revision surgery in patients with spinal fusion [56]. It has also been reported that spinal fusion patients are seven times more prone to THA dislocation [57]. An even more recent large meta-analysis including 10 studies which compared THA patients with or without previous surgery for spinal fusion (28.396 versus 1,550,291 respectively) [58]. The authors concluded that hip dislocations are more common in spinal fusion THA patients, revisions for any etiology, and overall complications.

The question of which surgery should be performed first, the spinal or hip arthroplasty, is of great importance. Malkani et al. [59] reported that patients (28.668) who underwent THA with pre-existing lumbar spinal fusion were at higher risk for dislocation and subsequent revision compared to patients (13.632) who had first the THA and then the lumbar spinal fusion, a finding that was also recapitulated in another same-year study [60].

At the same time, spinal pathology is associated with worse PROs [56,61,62]. Extreme vigilance should be exercised to identify patients with pre-existing spinal pathology, and a thorough workup should take place so that valid and safe preoperative planning to be conducted. Nowadays, sitting and standing radiographs of the pelvis and lumbar spine are used as screening tools for the detection of abnormal spinopelvic mobility [63].

Haffer et al. [64] identified the following steps as essential in evaluating the THA spinopelvic considerations in THA:

Examine the patient’s medical history and perform a clinical examination that importantly includes the detection of a possible hip flexion contracture.

Detection of symptoms related to pelvic and spinal pathologies.

Identification of individuals at risk of stiffness and sagittal imbalance of the spine.

Appropriate imaging, including anteroposterior standing and lateral standing radiographs of the pelvis. Additionally, sitting radiographs from L1 to the proximal femur should be performed.

Classification of the patients according to the mobility of spinopelvic junction and sagittal plane spinal balance.

Appropriate positioning of the acetabular cup according to preoperative classification.

A low threshold of using dual mobility acetabular implants in a high-risk patient.

To conclude, spinopelvic mobility is a crucial parameter that should be taken into account with preoperative evaluation and planning. The predicted increase in the incidence of patients with osteoarthritis and spinal pathologies, the anticipated increased numbers of THA and spinal correction surgery [40,65], and the evidence from contemporary meta-analyses [56,58] highlight the fact that surgeons should plan to account for biomechanical changes in spinopelvic junction and subsequent modified acetabular cup positioning, with a scope to avoid dislocations. The identification of high-risk patients should proceed through a detailed medical history, a meticulous physical examination, and diligent preoperative planning. Appropriate classification of spinopelvic mobility should guide the optimal intraoperative cup positioning.

## 5. Fast-Track Protocols in THA

The ‘Fast-track’ surgery concept was introduced almost twenty-five years ago and is generally defined as the coordinated peri-operative approach aiming at reducing surgical stress and facilitating post-operative recovery [66]. Despite the fact that the introduction of fast-track protocols have been introduced in orthopedics, their clinical implementation has mostly been slow [67]. The recent experience from their implementation in THA surgery has shown decreased hospitalization time reduction in post-operative length of stay (LOS), a shorter recovery time, and fast functional recovery without associated increases in morbidity and mortality [68,69]. Fast-track THA surgery units have grown with a well-defined organizational set-up assigned to provide a speedy peri-operative journey for the patients [70]. However, the benefits of fast-track THA for each patient taking into account the unique and different comorbidities of each individual is yet to be defined.

There is a generally accepted consensus that fast-track protocols in THA reduce hospitalization time, morbidity, and recovery time, without increasing readmission or compromising patients’ safety [71,72,73]. It is imperative to understand that despite the fact that many patients can be discharged very early after surgery (0–2 days), they have not fully recovered. In addition, widespread implementation of fast-track protocols is still not embodied by many institutions, with the length of stay being still around 4–6 days [74].

Pre-operative optimization is critical for the success of a fast-track protocols, since several factors can convey risk to the procedure [69]. These factors include smoking, alcohol consumption, psychiatric disease, polypharmacy, anemia, high BMI, and low physical activity, all of which could possibly prolong the LOS in the hospital. Lately, the value of correction of preexisting anemia has been emphasized as decreased hospitalization time and re-admission rates have been documented when this is preoperatively corrected and when minimal blood loss is intraoperatively practiced [75]. The role of postoperative anemia in the context of fast-track protocols is yet to be clearly defined.

The decrease in LOS has also been more recently confirmed. A large study by Grosso et al. [76] documented a statistically significant decrease in LOS and rates of 30-day morbidity following THA between 2006 and 2016. At the same time, Petersen et al. [77] studied LOS and morbidity after THA at nine high-volume orthopedic institutions that were practicing well-established fast-track THA protocols. The authors reported a continuous decrease in LOS to a median of one day and lower morbidity after THA. Berg et al. [78] studied the effect of fast-track programs on PROs, studying data from Swedish Knee and Hip Arthroplasty Registers. Minor differences in PROs favoring fast-track THA were reported. Despite the fact that these differences may not be clinically significant, the authors concluded that there are indications for the equal performance of fast tracts and conventional postoperative protocols.

Only sparse evidence is available with regard to the implementation of fast-track surgery on revision hip arthroplasty. Joseph et al., using data from 126 revision THAs from a single center, reported improvement in in-hospital outcomes by the implementation of fast-track/enhanced recovery protocols [79]. In particular, the improvement reported was evident in better pain management, less blood transfusion, and early functional recovery. A very recent study examined 1.345 hip arthroplasty revisions from 6 different centers [80]. They found that median LOS decreased from 6 days in 2010 to 2 days in 2018. However, the 90-day readmission rate showed a fluctuating and increasing trend from 13% in 2010 to 28% in 2018. The latter was attributed to preoperative factors such as a walking aid, psychiatric disorders, age ≥ 80 years, cardiac or pulmonary disease, and BMI ≥ 35 kg/m^2^. These findings suggest that for high-risk patients, an extended fast-track protocol should be implemented. This protocol should focus on appropriate optimization before surgery, technically sound surgery to avoid the risk of dislocation, and sufficient postoperative monitoring.

In conclusion, fast-track THA surgery is a valid option for non-outlier patients. Future research should emphasize adjusting the multiple domains of fast-track pathways. For the outlier and those with significant comorbidities, a modified protocol should be designed and implemented. Although fast-track protocols for THA have been present and successful for more than a decade, quality assurance data with continuous outcome monitoring of the LOS, complications, and readmissions are limited. This stands true for a number of parameters such as evidence-based analgesic techniques, rehabilitation protocols, blood management, and optimized organizational pathways.

## 6. Summary

Despite the well-evidenced fact that contemporary THA is an extremely safe and effective surgical operation, open-burning questions still pertain to various different domains. New technology and device-assisted arthroplasty, such as robotic and navigation technologies, enable personalized surgical treatment, partly by using safe zones of cup positioning based on pre-operative evaluations of spinopelvic mobility. However, they have not yet translated into improved early reported functional outcomes and their long-term outcomes remain to be evaluated. Spinopelvic mobility is a parameter that has largely been underestimated before and has come into play in recent years, and its use in THA planning is currently being investigated through multifaceted research. A stiff spine is particularly associated with significantly high dislocation risk. Findings from recent meta-analyses [56,58] suggest that meticulous preoperative planning taking into account spinopelvic mobility changes should be exercised by hip arthroplasty. Diligent preoperative evaluations, including appropriate imaging, should be used to identify patients at risk. It is crucial for the surgeon to identify those patients who are at increased risk for dislocation after total hip arthroplasty and personalize the treatment for these patients. Radiographs of the pelvis at standing and sitting positions are essential to measuring all the radiographic parameters to detect spinopelvic pathology. The two main surgical modification useful to mediate the high dislocation risk in these patients is the tailored acetabular cup position and the use of modern dual mobility cups. Finally, fast-track THA surgery protocols, although already available for over a decade, seem to work well only in the standard THA patient. Pain management is of paramount importance in early mobilization after total hip arthroplasty in fast-track protocols. Future studies should emphasize patients with special needs and/or a high co-morbidity burden and explore how the fast-track protocol can be implemented in such populations. THA patients’ needs and characteristics should always be carefully assessed while providing post-operative care and support.

## Figures and Tables

**Figure 1 medicina-59-00655-f001:**
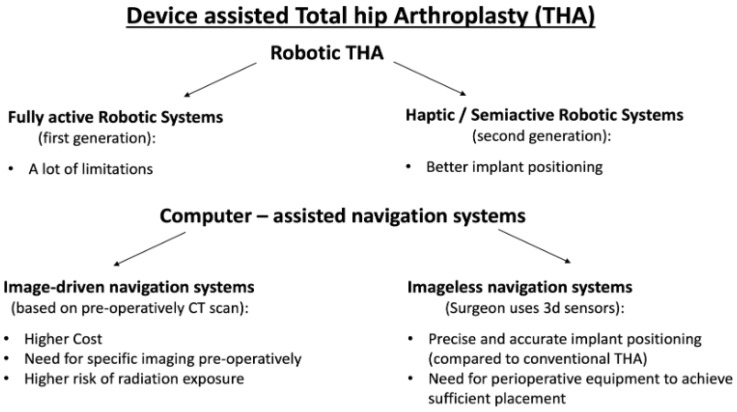
A summary of the major benefits and drawbacks of contemporary technologies used in THA.

## Data Availability

Not applicable.

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
