# Peer review of "The Top Three Burning Questions in Total Hip Arthroplasty"

_medicina, 2023, doi:10.3390/medicina59040655_

Round 1
Reviewer 1 Report
The paper needs minor spelling corrections.
A good up to date review
Author Response
We tried to correct the errors suggested by Reviewer 1.
Reviewer 2 Report
This is an interesting study, the authors were focused on the current three most debatable issues surrounding THA procedure. There are no methodological errors in the paper and there are some issues that need to be addressed in the current version of the manuscript. The following are the specific comments.
1. I wonder on what basis the author judges these three issues to be the most controversial? Whether there is support from references or were judged by the authors based on their own experiences?
2. I think it is necessary to explain different questions with appropriate diagrams.
3. The lack of line number affects the review of the manuscript.
Author Response
- This is a narrative review that was driven by the everyday practice of the authors. The three questions were arbitrarily chosen and there is no bibliographic and/or evidence-based selection of the questions. we have incorporated the phrase ". This is a narrative review that was driven by the everyday practise of the authors" into the manuscript.
- The comments and the evidence that is documented in the manuscript, are not based on a systematic review methodology or meta-analysis methodology.
- The line number has been incorporated as suggested